# Unsupervised Learning by Program Synthesis

**Kevin Ellis**
Department of Brain and Cognitive Sciences
Massachusetts Institute of Technology
ellisk@mit.edu

**Armando Solar-Lezama**
MIT CSAIL
Massachusetts Institute of Technology
asolar@csail.mit.edu

**Joshua B. Tenenbaum**
Department of Brain and Cognitive Sciences
Massachusetts Institute of Technology
jbt@mit.edu

## Abstract

We introduce an unsupervised learning algorithm that combines probabilistic modeling with solver-based techniques for program synthesis. We apply our techniques to both a visual learning domain and a language learning problem, showing that our algorithm can learn many visual concepts from only a few examples and that it can recover some English inflectional morphology. Taken together, these results give both a new approach to unsupervised learning of symbolic compositional structures, and a technique for applying program synthesis tools to noisy data.

## 1 Introduction

Unsupervised learning seeks to induce good latent representations of a data set. Nonparametric statistical approaches such as deep autoencoder networks, mixture-model density estimators, or nonlinear manifold learning algorithms have been very successful at learning representations of high-dimensional perceptual input. However, it is unclear how they would represent more abstract structures such as spatial relations in vision (e.g., *inside of* or *all in a line*) [2], or morphological rules in language (e.g., the different inflections of verbs) [1, 13]. Here we give an unsupervised learning algorithm that synthesizes programs from data, with the goal of learning such concepts. Our approach generalizes from small amounts of data, and produces interpretable symbolic representations parameterized by a human-readable programming language.

Programs (deterministic or probabilistic) are a natural knowledge representation for many domains [3], and the idea that inductive learning should be thought of as probabilistic inference over programs is at least 50 years old [6]. Recent work in learning programs has focused on supervised learning from noiseless input/output pairs, or from formal specifications [4]. Our goal here is to learn programs from noisy observations without explicit input/output examples. A central idea in unsupervised learning is compression: finding data representations that require the fewest bits to write down. We realize this by treating observed data as the output of an unknown program applied to unknown inputs. By doing joint inference over the program and the inputs, we recover compressive encodings of the observed data. The induced program gives a generative model for the data, and the induced inputs give an embedding for each data point.

Although a completely domain general method for program synthesis would be desirable, we believe this will remain intractable for the foreseeable future. Accordingly, our approach factors out the domain-specific components of problems in the form of a grammar for program hypotheses, and we show how this allows the same general-purpose tools to be used for unsupervised program synthesis in two very different domains. In a domain of visual concepts [5] designed to be natural for

humans but difficult for machines to learn, we show that our methods can synthesize simple graphics programs representing these visual concepts from only a few example images. These programs outperform both previous machine-learning baselines and several new baselines we introduce. We also study the domain of learning morphological rules in language, treating rules as programs and inflected verb forms as outputs. We show how to encode prior linguistic knowledge as a grammar over programs and recover human-readable linguistic rules, useful for both simple stemming tasks and for predicting the phonological form of new words.

## 2 The unsupervised program synthesis algorithm

The space of all programs is vast and often unamenable to the optimization methods used in much of machine learning. We extend two ideas from the program synthesis community to make search over programs tractable:

**Sketching:** In the sketching approach to program synthesis, one manually provides a *sketch* of the program to be induced, which specifies a rough outline of its structure [7]. Our sketches take the form of a probabilistic context-free grammar and make explicit the domain specific prior knowledge.

**Symbolic search:** Much progress has been made in the engineering of general-purpose symbolic solvers for Satisfiability Modulo Theories (SMT) problems [8]. We show how to translate our sketches into SMT problems. Program synthesis is then reduced to solving an SMT problem. These are intractable in general, but often solved efficiently in practice due to the highly constrained nature of program synthesis which these solvers can exploit.

Prior work on symbolic search from sketches has not had to cope with noisy observations or probabilities over the space of programs and inputs. Demonstrating how to do this efficiently is our main technical contribution.

### 2.1 Formalization as probabilistic inference

We formalize unsupervised program synthesis as Bayesian inference within the following generative model: Draw a program $f(\cdot)$ from a description length prior over programs, which depends upon the sketch. Draw $N$ inputs $\{I_i\}_{i=1}^N$ to the program $f(\cdot)$ from a domain-dependent description length prior $P_I(\cdot)$. These inputs are passed to the program to yield $\{z_i\}_{i=1}^N$ with $z_i \triangleq f(I_i)$ ($z_i$ "defined as" $f(I_i)$). Last, we compute the observed data $\{x_i\}_{i=1}^N$ by drawing from a noise model $P_{x|z}(\cdot|z_i)$.

Our objective is to estimate the unobserved $f(\cdot)$ and $\{I_i\}_{i=1}^N$ from the observed dataset $\{x_i\}_{i=1}^N$. We use this probabilistic model to define the description length below, which we seek to minimize:

$$\underbrace{-\log P_f(f)}_{\text{program length}} + \sum_{i=1}^N \Big( \underbrace{-\log P_{x|z}(x_i|f(I_i))}_{\text{data reconstruction error}} \quad \underbrace{-\log P_I(I_i)}_{\text{data encoding length}} \Big) \tag{1}$$

### 2.2 Defining a program space

We sketch a space of allowed programs by writing down a context free grammar $\mathcal{G}$, and write $\mathcal{L}$ to mean the set of all programs generated by $\mathcal{G}$. Placing uniform production probabilities over each non-terminal symbol in $\mathcal{G}$ gives a PCFG that serves as a prior over programs: the $P_f(\cdot)$ of Eq. 1. For example, a grammar over arithmetic expressions might contain rules that say: "expressions are either the sum of two expressions, or a real number, or an input variable $x$" which we write as

$$\mathcal{E} \to \mathcal{E} + \mathcal{E} \mid \mathbb{R} \mid x \tag{2}$$

Having specified a space of programs, we define the meaning of a program in terms of SMT primitives, which can include objects like tuples, real numbers, conditionals, booleans, etc [8]. We write $\tau$ to mean the set of expressions built of SMT primitives. Formally, we assume $\mathcal{G}$ comes equipped with a denotation for each rule, which we write as $[\![\cdot]\!] : \mathcal{L} \to \tau \to \tau$. The denotation of a rule in $\mathcal{G}$ is always written as a function of the denotations of that rule's children. For example, a denotation for the grammar in Eq. 2 is (where $I$ is a program input):

$$[\![\mathcal{E}_1 + \mathcal{E}_2]\!](I) = [\![\mathcal{E}_1]\!](I) + [\![\mathcal{E}_2]\!](I) \qquad [\![r \in \mathbb{R}]\!](I) = r \qquad [\![x]\!](I) = I \tag{3}$$

Defining the denotations for a grammar is straightforward and analogous to writing a "wrapper library" around the core primitives of the SMT solver. Our formalization factors out the grammar and the denotation, but they are tightly coupled and, in other synthesis tools, written down together [7, 9].

The denotation shows how to construct an SMT expression from a single program in $\mathcal{L}$, and we use it to build an SMT expression that represents the space of all programs such that its solution tells which program in the space solves the synthesis problem. The SMT solver then solves jointly for the program and its inputs, subject to an upper bound upon the total description length. This builds upon prior work in program synthesis, such as [9], but departs in the quantitative aspect of the constraints and in not knowing the program inputs. Due to space constraints, we only briefly describe the synthesis algorithm, leaving a detailed discussion to the Supplement.

We use Algorithm 1 to generate an SMT formula that (1) defines the space of programs $\mathcal{L}$; (2) computes the description length of a program; and (3) computes the output of a program on a given input. In Algorithm 1 the returned description length $l$ corresponds to the $-\log P_f(f)$ term of Eq. 1 while the returned evaluator $f(\cdot)$ gives us the $f(I_i)$ terms. The returned constraints $A$ ensure that the program computed by $f(\cdot)$ is a member of $\mathcal{L}$.

The SMT formula generated by Algorithm 1 must be supplemented with constraints that compute the data reconstruction error and data encoding length of Eq. 1. We handle infinitely recursive grammars by bounding the depth of recursive calls to the **Generate** procedure, as in [7].

SMT solvers are not designed to minimize loss functions, but to verify the satisfiability of a set of constraints. We minimize Eq. 1 by first asking the solver for any solution, then adding a constraint saying its solution must have smaller description length than the one found previously, etc. until it can find no better solution.

## 3 Experiments

### 3.1 Visual concept learning

Humans quickly learn new visual concepts, often from only a few examples [2, 5, 10]. In this section, we present evidence that an unsupervised program synthesis approach can also learn visual concepts from a small number of examples. Our approach is as follows: given a set of example images, we automatically parse them into a symbolic form. Then, we synthesize a program that maximally compresses these parses. Intuitively, this program encodes the common structure needed to draw each of the example images.

---

**Algorithm 1** SMT encoding of programs generated by production $P$ of grammar $\mathcal{G}$

**function Generate($\mathcal{G}$,$[\![\cdot]\!]$,$P$):**
**Input:** Grammar $\mathcal{G}$, denotation $[\![\cdot]\!]$, non-terminal $P$
**Output:** Description length $l : \tau$,
$\quad\quad$ evaluator $f : \tau \to \tau$, assertions $A : 2^\tau$
choices $\leftarrow \{P \to K(P', P'', \ldots) \in \mathcal{G}\}$
$n \leftarrow |\text{choices}|$
**for** $r = 1$ **to** $n$ **do**
$\quad$ **let** $K(P_r^1, \ldots, P_r^k) = \text{choices}(r)$
$\quad$ **for** $j = 1$ **to** $k$ **do**
$\quad\quad$ $l_r^j, f_r^j, A_r^j \leftarrow$ **Generate($\mathcal{G}$,$[\![\cdot]\!]$,$P_r^j$)**
$\quad$ **end for**
$\quad$ $l_r \leftarrow \sum_j l_r^j$
$\quad$ // Denotation is a function of child denotations
$\quad$ // Let $g_r$ be that function for choices$(r)$
$\quad$ // $Q^1, \cdots, Q^k : \mathcal{L}$ are arguments to constructor $K$
$\quad$ **let** $g_r([\![Q^1]\!](I), \cdots, [\![Q^k]\!](I)) =$
$\quad\quad\quad\quad [\![K(Q^1, \ldots, Q^k)]\!](I)$
$\quad$ $f_r(I) \leftarrow g_r(f_r^1(I), \cdots, f_r^k(I))$
**end for**
// Indicator variables specifying which rule is used
// Fresh variables unused in any existing formula
$c_1, \cdots, c_n = $ **FreshBooleanVariable()**
$A_1 \leftarrow \bigvee_j c_j$
$A_2 \leftarrow \forall j \neq k : \neg(c_j \wedge c_k)$
$A \leftarrow A_1 \cup A_2 \cup \bigcup_{r,j} A_r^j$
$l = \log n + \text{if}(c_1, l_1, \text{if}(c_2, l_2, \cdots))$
$f(I) = \text{if}(c_1, f_1(I), \text{if}(c_2, f_2(I), \cdots))$
**return** $l, f, A$

---

We take our visual concepts from the Synthetic Visual Reasoning Test (SVRT), a set of visual classification problems which are easily parsed into distinct shapes. Fig. 1 shows three examples of SVRT concepts. Fig. 2 diagrams the parsing procedure for another visual concept: two arbitrary shapes bordering each other.

We defined a space of simple graphics programs that control a turtle [11] and whose primitives include rotations, forward movement, rescaling of shapes, etc.; see Table 1. Both the learner's observations and the graphics program outputs are image parses, which have three sections: (1) A list of shapes. Each shape is a tuple of a unique ID, a scale from 0 to 1, and $x, y$ coordinates:

$\langle \mathrm{id}, \mathrm{scale}, x, y \rangle$. (2) A list of containment relations $\mathrm{contains}(i, j)$ where $i, j$ range from one to the number of shapes in the parse. (3) A list of reflexive borders relations $\mathrm{borders}(i, j)$ where $i, j$ range from one to the number of shapes in the parse.

The algorithm in Section 2.2 describes purely functional programs (programs without state), but the grammar in Table 1 contains imperative commands that modify a turtle's state. We can think of imperative programs as syntactic sugar for purely functional programs that pass around a state variable, as is common in the programming languages literature [7].

The grammar of Table 1 leaves unspecified the number of program inputs. When synthesizing a program from example images, we perform a grid search over the number of inputs. Given images with $N$ shapes and maximum shape ID $D$, the grid search considers $D$ input shapes, 1 to $N$ input positions, 0 to 2 input lengths and angles, and 0 to 1 input scales. We set the number of imperative draw commands (resp. borders, contains) to $N$ (resp. number of topological relations).

We now define a noise model $P_{x|z}(\cdot|\cdot)$ that specifies how a program output $z$ produces a parse $x$, by defining a procedure for sampling $x$ given $z$. First, the $x$ and $y$ coordinates of each shape are perturbed by additive noise drawn uniformly from $-\delta$ to $\delta$; in our experiments, we put $\delta = 3$. Then, optional borders and contains relations (see Table 1) are erased with probability 1/2. Last, because the order of the shapes is unidentifiable, both the list of shapes and the indices of the borders/containment relations are randomly permuted. The Supplement has the SMT encoding of the noise model and priors over program inputs, which are uniform.

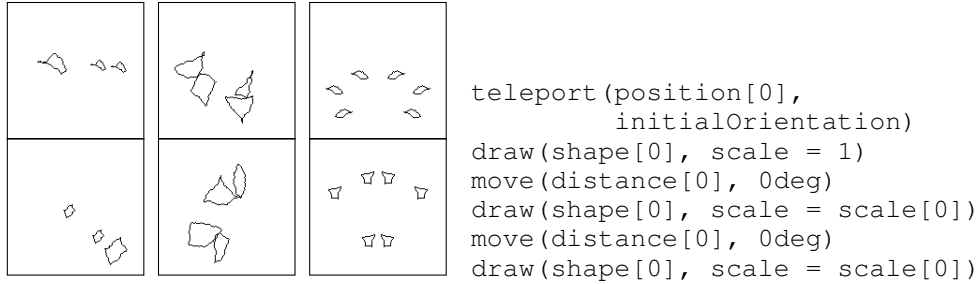

```
teleport(position[0],
         initialOrientation)
draw(shape[0], scale = 1)
move(distance[0], 0deg)
draw(shape[0], scale = scale[0])
move(distance[0], 0deg)
draw(shape[0], scale = scale[0])
```

Figure 1: Left: Pairs of examples of three SVRT concepts taken from [5]. Right: the program we synthesize from the leftmost pair. This is a turtle program capable of drawing this pair of pictures and is parameterized by a set of latent variables: shape, distance, scale, initial position, initial orientation.

To encourage translational and rotational invariance, the first turtle command is constrained to always be a teleport to a new location, and the initial orientation of the turtle, which we write as $\theta_0$, is made an input to the synthesized graphics program.

We are introducing an unsupervised learning algorithm, but the SVRT consists of supervised binary classification problems. So we chose to evaluate our visual concept learner by having it solve these classification problems. Given a test image $t$ and a set of examples $E_1$ (resp. $E_2$) from class $C_1$ (resp. $C_2$), we use the decision rule $P(t|E_1) \gtrless_{C_2}^{C_1} P(t|E_2)$, or equivalently $P_x(\{t\} \cup E_1) P_x(E_2) \gtrless_{C_2}^{C_1} P_x(E_1) P_x(\{t\} \cup E_2)$. Each term in this decision rule is written as a marginal probability, and we approximate each marginal by lower bounding it by the largest term in its corresponding sum. This gives

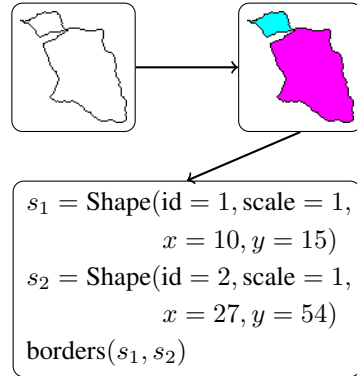

Figure 2: The parser segments shapes and identifies their topological relations (contains, borders), emmitting their coordinates, topological relations, and scales.

$$\underbrace{-l(\{t\} \cup E_1)}_{\approx \log P_x(\{t\} \cup E_1)} \quad \underbrace{-l(E_2)}_{\approx \log P_x(E_2)} \quad \overset{C_1}{\underset{C_2}{\gtrless}} \quad \underbrace{-l(E_1)}_{\approx \log P_x(E_1)} \quad \underbrace{-l(\{t\} \cup E_2)}_{\approx \log P_x(\{t\} \cup E_2)} \tag{4}$$

| Grammar rule | English description |
|---|---|
| $\mathcal{E} \to (\mathcal{M}; \mathcal{D})^+; \mathcal{C}^+; \mathcal{B}^+$ | Alternate move/draw; containment relations; borders relations |
| $\mathcal{M} \to \texttt{teleport}(\mathcal{R}, \theta_0)$ | Move turtle to new location $\mathcal{R}$, reset orientation to $\theta_0$ |
| $\mathcal{M} \to \texttt{move}(\mathcal{L}, \mathcal{A})$ | Rotate by angle $\mathcal{A}$, go forward by distance $\mathcal{L}$ |
| $\mathcal{M} \to \texttt{flipX()}|\texttt{flipY()}$ | Flip turtle over X/Y axis |
| $\mathcal{M} \to \texttt{jitter()}$ | Small perturbation to turtle position |
| $\mathcal{D} \to \texttt{draw}(\mathcal{S}, \mathcal{Z})$ | Draw shape $\mathcal{S}$ at scale $\mathcal{Z}$ |
| $\mathcal{Z} \to 1|z_1|z_2|\cdots$ | Scale is either 1 (no rescaling) or program input $z_j$ |
| $\mathcal{A} \to 0°| \pm 90°|\theta_1|\theta_2|\cdots$ | Angle is either $0°$, $\pm 90°$, or a program input $\theta_j$ |
| $\mathcal{R} \to r_1|r_2|\cdots$ | Positions are program inputs $r_j$ |
| $\mathcal{S} \to s_1|s_2|\cdots$ | Shapes are program inputs $s_j$ |
| $\mathcal{L} \to \ell_1|\ell_2|\cdots$ | Lengths are program inputs $\ell_j$ |
| $\mathcal{C} \to \texttt{contains}(\mathbb{Z}, \mathbb{Z})$ | Containment between integer indices into drawn shapes |
| $\mathcal{C} \to \texttt{contains?}(\mathbb{Z}, \mathbb{Z})$ | Optional containment between integer indices into drawn shapes |
| $\mathcal{B} \to \texttt{borders}(\mathbb{Z}, \mathbb{Z})$ | Bordering between integer indices into drawn shapes |
| $\mathcal{B} \to \texttt{borders?}(\mathbb{Z}, \mathbb{Z})$ | Optional bordering between integer indices into drawn shapes |

Table 1: Grammar for the vision domain. The non-terminal $\mathcal{E}$ is the start symbol for the grammar. The token ; indicates sequencing of imperative commands. Optional bordering/containment holds with probability $1/2$. See the Supplement for denotations of each grammar rule.

where $l(\cdot)$ is

$$l(E) \triangleq \min_{f, \{I_e\}_{e \in E}} - \log P_f(f) - \left( \sum_{e \in E} \log P_I(I_e) + \log P_{x|z}(E_e|f(I_e)) \right) \qquad (5)$$

So, we induce 4 programs that maximally compress a different set of image parses: $E_1$, $E_2$, $E_1 \cup \{t\}$, $E_2 \cup \{t\}$. The maximally compressive program is found by minimizing Eq. 5, putting the observations $\{x_i\}$ as the image parses, putting the inputs $\{I_e\}$ as the parameters of the graphics program, and generating the program $f(\cdot)$ by passing the grammar of Table 1 to Algorithm 1.

We evaluated the classification accuracy across each of the 23 SVRT problems by sampling three positive and negative examples from each class, and then evaluating the accuracy on a held out test example. 20 such estimates were made for each problem. We compare with three baselines, as shown in Fig. 3. (1) To control for the effect of our parser, we consider how well discriminative classification on the image parses performs. For each image parse, we extracted the following features: number of distinct shapes, number of rescaled shapes, and number of containment/bordering relations, for 4 integer valued features. Following [5] we used Adaboost with decision stumps on these parse features. (2) We trained two convolutional network architectures for each SVRT problem, and found that a variant of LeNet5 [12] did best; we report those results here. The Supplement has the network parameters and results for both architectures. (3) In [5] several discriminative baselines are introduced. These models are trained on low-level image features; we compare with their best-performing model, which fed 10000 examples to Adaboost with decision stumps. Unsupervised program synthesis does best in terms of average classification accuracy, number of SVRT problems solved at $\geq 90\%$ accuracy,[1] and correlation with the human data.

We do not claim to have solved the SVRT. For example, our representation does not model some geometric transformations needed for some of the concepts, such as rotations of shapes. Additionally, our parsing procedure occasionally makes mistakes, which accounts for the many tasks we solve at accuracies between 90% and 100%.

## 3.2 Morphological rule learning

How might a language learner discover the rules that inflect verbs? We focus on English inflectional morphology, a system with a long history of computational modeling [13]. Viewed as an unsupervised learning problem, our objective is to find a compressive representation of English verbs.

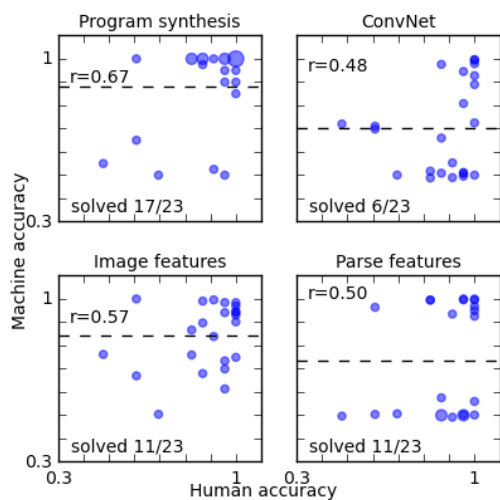

Figure 3: Comparing human performance on the SVRT with classification accuracy for machine learning approaches. Human accuracy is the fraction of humans that learned the concept: 0% is chance level. Machine accuracy is the fraction of correctly classified held out examples: 50% is chance level. Area of circles is proportional to the number of observations at that point. Dashed line is average accuracy. Program synthesis: this work trained on 6 examples. ConvNet: A variant of LeNet5 trained on 2000 examples. Parse (Image) features: discriminative learners on features of parse (pixels) trained on 6 (10000) examples. Humans given an average of 6.27 examples and solve an average of 19.85 problems [5].

We make the following simplification: our learner is presented with triples of $\langle\text{lexeme, tense, word}\rangle^2$. This ignores many of the difficulties involved in language acquisition, but see [14] for a unsupervised approach to extracting similar information from corpora. We can think of these triples as the entries of a matrix whose columns correspond to different tenses and whose rows correspond to different lexemes; see Table 3. We regard each row of this matrix as an observation (the $\{x_i\}$ of Eq. 1) and identify stems with the inputs to the program we are to synthesize (the $\{I_i\}$ of Eq. 1). Thus, our objective is to synthesize a program that maps a stem to a tuple of inflections. We put a description length prior over the stem and detail its SMT encoding in the the Supplement. We represent words as sequences of phonemes, and define a space of programs that operate upon words, given in Table 2.

English inflectional verb morphology has a set of regular rules that apply for almost all words, as well as a small set of words whose inflections do not follow a regular rule: the "irregular" forms. We roll these irregular forms into the noise model: with some small probability $\epsilon$, an inflected form is produced not by applying a rule to the stem, but by drawing a sequence of phonemes from a description length prior. In our experiments, we put $\epsilon = 0.1$. This corresponds to a simple "rules plus lexicon" model of morphology, which is oversimplified in many respects but has been proposed in the past as a crude approximation to the actual system of English morphology [13]. See the Supplement for the SMT encoding of our noise model.

In conclusion, the learning problem is as follows: given triples of $\langle\text{lexeme, tense, word}\rangle$, jointly infer the regular rules, the stems, and which words are irregular exceptions.

We took five inflected forms of the top 5000 lexemes as measured by token frequency in the CELEX lexical inventory [15]. We split this in half to give 2500 lexemes for training and testing, and trained our model using Random Sample Consensus (RANSAC) [16]. Concretely, we sampled many subsets of the data, each with 4, 5, 6, or 7 lexemes (thus 20, 25, 30, or 35 words), and synthesized the program for each subset minimizing Eq. 1. We then took the program whose likelihood on the training set was highest. Fig. 4 plots the likelihood on the testing set as a function of the number of subsets (RANSAC iterations) and the size of the subsets (# of lexemes). Fig. 5 shows the program that assigned the highest likelihood to the training data; it also had the highest likelihood on the testing data. With 7 lexemes, the learner consistently recovers the regular linguistic rule, but with less data, it recovers rules that are almost as good, degrading more as it receives less data.

Most prior work on morphological rule learning falls into two regimes: (1) supervised learning of the phonological form of morphological rules; and (2) unsupervised learning of morphemes from corpora. Because we learn from the lexicon, our model is intermediate in terms of supervision. We compare with representative systems from both regimes as follows:

| Grammar rule | English description |
|---|---|
| $\mathcal{E} \rightarrow \langle \mathcal{C}, \cdots, \mathcal{C} \rangle$ | Programs are tuples of conditionals, one for each tense |
| $\mathcal{C} \rightarrow \mathcal{R} \vert$ if $(\mathcal{G})$ $\mathcal{R}$ else $\mathcal{C}$ | Conditionals have return value $\mathcal{R}$, guard $\mathcal{G}$, else condition $\mathcal{C}$ |
| $\mathcal{R} \rightarrow \texttt{stem} + \texttt{phoneme}^*$ | Return values append a suffix to a stem |
| $\mathcal{G} \rightarrow [\mathcal{V}\mathcal{P}\mathcal{M}\mathcal{S}]$ | Guards condition upon voicing, manner, place, sibilancy |
| $\mathcal{V} \rightarrow \mathcal{V}' \vert ?$ | Voicing specifies of voice $\mathcal{V}'$ or doesn't care |
| $\mathcal{V}' \rightarrow \texttt{VOICED} \vert \texttt{UNVOICED}$ | Voicing options |
| $\mathcal{P} \rightarrow \mathcal{P}' \vert ?$ | Place specifies a place of articulation $\mathcal{P}'$ or doesn't care |
| $\mathcal{P}' \rightarrow \texttt{LABIAL} \vert \cdots$ | Place of articulation features |
| $\mathcal{M} \rightarrow \mathcal{M}' \vert ?$ | Manner specifies a manner of articulation $\mathcal{M}'$ or doesn't care |
| $\mathcal{M}' \rightarrow \texttt{FRICATIVE} \vert \cdots$ | Manner of articulation features |
| $\mathcal{S} \rightarrow \mathcal{S}' \vert ?$ | Sibilancy specifies a sibilancy $\mathcal{S}'$ or doesn't care |
| $\mathcal{S}' \rightarrow \texttt{SIBILANT} \vert \texttt{NOTSIBIL}$ | Sibilancy is a binary feature |

Table 2: Grammar for the morphology domain. The non-terminal $\mathcal{E}$ is the start symbol for the grammar. Each guard $\mathcal{G}$ conditions on phonological properties of the end of the stem: voicing, place, manner, and sibilancy. Sequences of phonemes are encoded as tuples of $\langle \text{length}, \text{phoneme}_1, \text{phoneme}_2, \cdots \rangle$. See the Supplement for denotations of each grammar rule.

| Lexeme | Present | Past | 3rd Sing. Pres. | Past Part. | Prog. |
|---|---|---|---|---|---|
| style | staɪl | staɪld | staɪlz | staɪld | staɪlɪŋ |
| run | rʌn | ræn | rʌnz | rʌn | rʌnɪŋ |
| subscribe | səbskraɪb | səbskraɪbd | səbskraɪbz | səbskraɪbd | səbskraɪbɪŋ |
| rack | ræk | rækt | ræks | rækt | rækɪŋ |

Table 3: Example input to the morphological rule learner

The Morfessor system [17] induces morphemes from corpora which it then uses for segmentation. We used Morfessor to segment phonetic forms of the inflections of our 5000 lexemes; compared to the ground truth inflection transforms provided by CELEX, it has an error rate of 16.43%. Our model segments the same verbs with an error rate of 3.16%. This experiment is best seen as a sanity check: because our system knows a priori to expect only suffixes and knows which words must share the same stem, we expect better performance due to our restricted hypothesis space. To be clear, we are not claiming that we have introduced a stemmer that exceeds or even meets the state-of-the-art.

In [1] Albright and Hayes introduce a supervised morphological rule learner that induces phonological rules from examples of a stem being transformed into its inflected form. Because our model learns a joint distribution over all of the inflected forms of a lexeme, we can use it to predict inflections conditioned upon their present tense. Our model recovers the regular inflections, but does not recover the so-called "islands of reliability" modeled in [1]; e.g., our model predicts that the past tense of the nonce word *glee* is *gleed*, but does not predict that a plausible alternative past tense is *gled*, which the model of Albright and Hayes does. This deficiency is because the space of programs in Table 2 lacks the ability to express this class of rules.

## 4 Discussion

### 4.1 Related Work

Inductive programming systems have a long and rich history [4]. Often these systems use stochastic search algorithms, such as genetic programming [18] or MCMC [19]. Others sufficiently constrain the hypothesis space to enable fast exact inference [20]. The inductive logic programming community has had some success inducing Prolog programs using heuristic search [4]. Our work is motivated by the recent successes of systems that put program synthesis in a probabilistic framework [21, 22]. The program synthesis community introduced solver-based methods for learning programs [7, 23, 9], and our work builds upon their techniques.

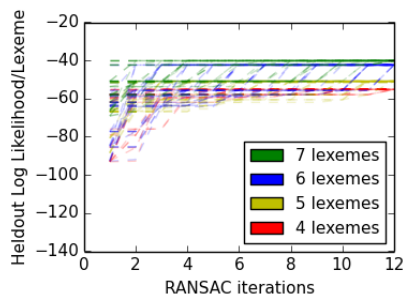

Figure 4: Learning curves for our morphology model trained using RANSAC. At each iteration, we sample 4, 5, 6, or 7 lexemes from the training data, fit a model using their inflections, and keep the model if it has higher likelihood on the training data than other models found so far. Each line was run on a different permutation of the samples.

```
PRESENT  = stem
PAST     = if [ CORONAL STOP ]
              stem + ɪd
           if [ VOICED ]
              stem + d
           else
              stem + t
PROG.    = stem + ɪŋ
3rdSing  = if [ SIBILANT ]
              stem + ɪz
           if [ VOICED ]
              stem + z
           else
              stem + s
```

Figure 5: Program synthesized by morphology learner. Past Participle program was the same as past tense program.

There is a vast literature on computational models of morphology. These include systems that learn the phonological form of morphological rules [1, 13, 24], systems that induce morphemes from corpora [17, 25], and systems that learn the productivity of different rules [26]. In using a general framework, our model is similar in spirit to the early connectionist accounts [24], but our use of symbolic representations is more in line with accounts proposed by linguists, like [1].

Our model of visual concept learning is similar to inverse graphics, but the emphasis upon synthesizing programs is more closely aligned with [2].We acknowledge that convolutional networks are engineered to solve classification problems qualitatively different from the SVRT, and that one could design better neural network architectures for these problems. For example, it would be interesting to see how the very recent DRAW network [27] performs on the SVRT.

## 4.2 A limitation of the approach: Large datasets

Synthesizing programs from large datasets is difficult, and complete symbolic solvers often do not degrade gracefully as the problem size increases. Our morphology learner uses RANSAC to sidestep this limitation, but we anticipate domains for which this technique will be insufficient. Prior work in program synthesis introduced Counter Example Guided Inductive Synthesis (CEGIS) [7] for learning from a large or possibly infinite family of examples, but it cannot accomodate noise in the data. We suspect that a hypothetical RANSAC/CEGIS hybrid would scale to large, noisy training sets.

## 4.3 Future Work

The two key ideas in this work are (1) the encoding of soft probabilistic constraints as hard constraints for symbolic search, and (2) crafting a domain specific grammar that serves both to guide the symbolic search and to provide a good inductive bias. Without a strong inductive bias, one cannot possibly generalize from a small number of examples. Yet humans can, and AI systems should, learn over time what constitutes a good prior, hypothesis space, or sketch. Learning a good inductive bias, as done in [22], and then providing that inductive bias to a solver, may be a way of advancing program synthesis as a technology for artificial intelligence.

## Acknowledgments

We are grateful for discussions with Timothy O'Donnell on morphological rule learners, for advice from Brendan Lake and Tejas Kulkarni on the convolutional network baselines, and for the suggestions of our anonymous reviewers. This material is based upon work supported by funding from NSF award SHF-1161775, from the Center for Minds, Brains and Machines (CBMM) funded by NSF STC award CCF-1231216, and from ARO MURI contract W911NF-08-1-0242.

## Footnotes

[1]Humans "learn the task" after seven consecutive correct classifications [5]. Seven correct classifications are likely to occur when classification accuracy is $\geq 0.5^{1/7} \approx 0.9$

[2]The lexeme is the meaning of the stem or root; for example, *run*, *ran*, *runs* all share the same lexeme

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
