[Supplementary Material]

# Supplement for: Unsupervised Learning by Program Synthesis

Kevin Ellis
Department of Brain and Cognitive Sciences
Massachusetts Institute of Technology
ellisk@mit.edu

Armando Solar-Lezama
MIT CSAIL
Massachusetts Institute of Technology
asolar@csail.mit.edu

Joshua B. Tenenbaum
Department of Brain and Cognitive Sciences
Massachusetts Institute of Technology
jbt@mit.edu

## 1 The synthesis algorithm

Unsupervised program synthesis is a domain-general framework for defining domain-specific program synthesis systems. For each domain, we expect the user to sketch a space of program hypotheses. For example, in a domain of regression problems the space of programs might include piecewise polynomials, and in a domain of visual concepts the space of programs might include graphics primitives. As part of the probabilistic framing of unsupervised program synthesis, the user must also write down a (prior) probability distribution over program inputs.

Given the program sketch and prior program input probabilities, we give a domain-general algorithm that inductively synthesizes programs from noisy data sets where we have a model of the noise process. The general idea is to compile both the soft constraints of the probabilistic models and the hard constraints of the program space into a set of equations that an SMT solver can jointly reason over. Section 1.1 gives a heuristic overview of this compilation algorithm, while Section 1.2 formalizes the unsupervised program synthesis algorithm.

### 1.1 A motivating example

We consider spaces of programs defined by a context free grammar, $\mathcal{G}$. As a running example for this section, we will consider the space of programs generated by the grammar

$$E \rightarrow E + E \mid \mathbb{R} \mid x \tag{1}$$

We can think of each program in this space as being a path through an AND/OR graph. Each OR node corresponds to a choice between the three productions $+$, $\mathbb{R}$, $x$, and each AND node corresponds to generating the descendents of a production. In our running example, the AND nodes have two descendents and correspond to the two summands for the $+$ production.

As in prior program synthesis work [1], we propositionalize, or "ground out," the infinite space of programs generated by $\mathcal{G}$. For each OR node, we introduce mutually exclusive binary indicator variables, one for each descendent, that indicate which of the productions was chosen at that OR node. These binary indicator variables, written as $c_i^j$, are best thought of as "control bits" that specify the structure of the program. Crucially, each assignment to these variables picks out exactly one program. See Figure 1.

These binary indicator variables, along with the mutual exclusivity constraints, constrain the structure, or syntax, of the programs. We also need to introduce constraints that give a semantics to the program space. In other words, these constraints describe the relationship between the inputs to the program and its outputs.

We run the program by recursively computing the value of each node in the program's syntax tree from the value of its immediate descendents. This computation depends upon the structure of the program, and

Figure 1: Defining the space of programs generated by the grammar in Equation 1 and modeling their execution on an input variable $x$. Ellipses indicate elided subtrees or constraints. Not shown: constraints responsible for description length calculations.

hence the values of the $c_i^j$. Using the example in Figure 1: if $c_3^1$ holds, then the $+$ production was used, and so $E_1$ node is equal to the sum of the $E_2$ and $E_3$ nodes. Observe that we need to introduce constraints separately for each input $x$ to the program that model the execution of the program on $x$.

The description length of the program specified by the $c_i^j$ variables is similarly computed recursively from the description length of its subprograms. Our probabilistic model assumes equally likely production rules, and so the description length of a program (an OR node in Figure 1) is just the logarithm of the number of choices in the grammar, plus the description length of any subtrees. For example, the description length of the program rooted at $E_1$, which we write $\ell(E_1)$, is

$$\ell(E_1) = \underbrace{\log 3}_{\log(\# \text{ choices})} + c_1^1 \underbrace{\ell(\mathbb{R}_1)}_{\text{prior over } \mathbb{R}} + c_2^1 \underbrace{\ell(x)}_{=0} + c_3^1(\ell(E_2) + \ell(E_3)) \tag{2}$$

Up until now we have just described a framework for writing down spaces of programs and modeling their execution, alongside a description length calculation. In unsupervised program synthesis, we observe noisy program outputs, which we will write as $\{x_i\}_{i=1}^N$. The program inputs are treated as latent variables, which we will write as $\{I_i\}_{i=1}^N$. We assume a space of programs generated as in Figure 1, and write $E_1(I_i)$ to mean the value of the program rooted at $E_1$ evaluated at $I_i$. Then our total description length is

$$\ell(E_1) - \sum_{i=1}^N \log P(x_i | E_1(I_i)) - \sum_{i=1}^N \log P(I_i) \tag{3}$$

subject to the constraints calculating description lengths (eg, Equation 2), constraining the space of programs (eg, $c_i^j$'s in Figure 1) and evaluating the program on each $I_i$ (eg, the implications in Figure 1). We assume a form for $P(x_i | E_1(x_i))$ and $P(I_i)$ amenable to manipulation by the constraint solver: specifically, a form that can be written down as an SMT term.

## 1.2 The algorithm in detail

Let $\mathcal{G}$ be a context free grammar specified as a set of rules of the form

$$N \rightarrow K(N', N'', \cdots) \tag{4}$$

where $N, N', N'', \cdots$ are non-terminal symbols in the grammar and $K$ is a unique token, referred to as the *constructor* of a particular grammar rule. Let $\mathcal{L}$ be the set of all strings generated by the grammar $\mathcal{G}$.

Our constraints will take the form of expressions written in terms of SMT primitives, called *terms* within the SMT solving community. We refer the reader to [2, 3] for overviews of SMT syntax and Z3 in particular, the state-of-the-art SMT solver we use in this work. Let $\tau$ be the set of all SMT terms. We will write write out members of $\tau$ in typewriter font, so for instance `(ite (not (<= 0 x)) (+ 1 2) (f x))` and `(let ((x 0) (y 5)) (exp (- x y)))` are both SMT terms.

We assume that the user specifies a denotation, written $\llbracket \cdot \rrbracket$, that tells how to evaluate programs drawn from $\mathcal{G}$. The denotation maps a program and an input to the output of that program run on that input, so $\llbracket \cdot \rrbracket$ has the type $\mathcal{L} \to \tau \to \tau$. We further restrict ourselves to program denotations written only as functions of the denotations of the program's immediate subprograms. This is a natural condition satisfied by functional programming languages. For example, a denotation for the grammar in Equation 1 is

$$\llbracket E_1 + E_2 \rrbracket(\texttt{x}) = \texttt{(+ } \llbracket E_1 \rrbracket(\texttt{x}) \text{ } \llbracket E_2 \rrbracket(\texttt{x})\texttt{)} \qquad \llbracket r \in \mathbb{R} \rrbracket(\texttt{x}) = r \qquad \llbracket x \rrbracket(\texttt{x}) = \texttt{x} \tag{5}$$

The program synthesis problem also includes latent program inputs $\{I_i\}_{i=1}^N$, each of which is an SMT term (their exact form is domain-specific). Writing $f(\cdot)$ for the latent program, our objective now is to write down an algorithm that

- Defines the space of programs generated by $\mathcal{G}$ using the "control bits" idea of Section 1.1

- Evaluates $z_i \triangleq f(I_i)$ ($z_i$ "defined as" $f(I_i)$) given an assignment to the "control bits"

- Calculates the description length of the program specified by the "control bits"

The idea is to recursively walk the AND/OR graph generated by $\mathcal{G}$, which Algorithm 1 does.

Now that we have a space of programs evaluated on $N$ inputs, we need to connect these objects to the observed dataset, $\{x_i\}_{i=1}^N$. This happens through the description length we seek to minimize:

$$\underbrace{-\log P_f(f)}_{\text{program length}} + \sum_{i=1}^N \left( \underbrace{-\log P_{x|z}(x_i|f(I_i))}_{\text{data reconstruction error}} \quad \underbrace{-\log P_I(I_i)}_{\text{data encoding length}} \right) \tag{6}$$

We assume that the prior over program inputs is written down as a function $p(\cdot) : \tau \to \tau$ that maps a program input $I$ to its negative log probability, $-\log P_I(I)$. Similarly the noise model is written down as a function $n(\cdot|\cdot) : \tau^2 \to \tau$ that maps an observation $x$ and a program output $z$ to the negative log probability, $-\log P_{x|z}(x|z)$.

We minimize Equation 6 by first asking the solver for any solution to the constraints on the program space, then adding a constraint saying its solution must have smaller description length than the one found previously, etc. until it proves no better solution exists. Algorithm 2 implements this iterative optimization procedure.

**Algorithm 1** SMT encoding of programs generated by production $P$ of grammar $\mathcal{G}$

---

**function Generate($\mathcal{G}$,$[\![\cdot]\!]$,$P$,$\{I_i\}_{i=1}^N$):**
**Input:** Grammar $\mathcal{G}$, denotation $[\![\cdot]\!]$, non-terminal $P$, inputs $\{I_i\}_{i=1}^N$
**Output:** Description length $l : \tau$,
        outputs $\{z_i\}_{i=1}^N$, constraints $A : 2^\tau$
choices $\leftarrow \{P \rightarrow K(P', P'', \ldots) \in \mathcal{G}\}$
$n \leftarrow |\text{choices}|$
**for** $r = 1$ **to** $n$ **do**
  **let** $K(P_r^1, \ldots, P_r^k) = \text{choices}(r)$
  **for** $j = 1$ **to** $k$ **do**
    // Variable $z_r^{j,i}$ is output of $j^{th}$ child of $r^{th}$ rule choice on $i^{th}$ input
    $l_r^j, \{z_r^{j,i}\}_{i=1}^N, A_r^j \leftarrow$ **Generate($\mathcal{G}$,$[\![\cdot]\!]$,$P_r^j$,$\{I_i\}_{i=1}^N$)**
  **end for**
  $l_r \leftarrow$ (+ $l_r^1$ $l_r^2$ $\cdots$ $l_r^k$)
  // Denotation is a function of child denotations and input $I$
  // Let $g_r$ be that function for choices($r$)
  // $Q^1, \cdots, Q^k : \mathcal{L}$ are arguments to the constructor $K$
  **let** $g_r([\![Q^1]\!](I), \cdots, [\![Q^k]\!](I), I) =$
                $[\![K(Q^1, \ldots, Q^k)]\!](I)$
  **for** $i = 1$ **to** $N$ **do**
    $z_r^i \leftarrow g_r(z_r^{1,i}, \cdots, z_r^{k,i}, I_i)$
  **end for**
**end for**
// Indicator variables specifying which rule is used
// Fresh variables unused in any existing formula
$c_1, \cdots, c_n =$ **FreshBooleanVariable()**
$A_1 \leftarrow \bigvee_j c_j$
$A_2 \leftarrow \forall j \neq k : \neg(c_j \wedge c_k)$
$A \leftarrow A_1 \cup A_2 \cup \bigcup_{r,j} A_r^j$
// (ite . . .) is the SMT term for an if-then-else, or ternary, operator
$l \leftarrow$ (+ $\log n$ (ite $c_1$ $l_1$ (ite $c_2$ $l_2$ $\cdots$ )))
**for** $i = 1$ **to** $N$ **do**
  $z_i \leftarrow$ (ite $c_1$ $z_1^i$ (ite $c_2$ $z_2^i$ $\cdots$ ))
**end for**
**return** $l, \{z_i\}_{i=1}^N, A$

---

---

**Algorithm 2** The unsupervised program synthesis algorithm

---
 **function UnsupervisedProgramSynthesis($\mathcal{G}$,$P$,$[\![\cdot]\!]$,$\{x_i\}_{i=1}^N$, $n(\cdot|\cdot)$,$p(\cdot)$):**
 **Input:** Grammar $\mathcal{G}$, grammar start symbol $P$, denotation $[\![\cdot]\!]$, observations $\{x_i\}_{i=1}^N$, noise model $n(\cdot|\cdot)$, input model $p(\cdot)$
 **Output:** Program $f : \mathcal{L}$, $N$ program inputs, description length $\ell : \mathbb{R}$
 // $I_i : \tau$ is an (unknown) program input the SMT solver will solve for
 // Fresh variables unused in any existing formula
 $I_1, I_2, \cdots, I_N =$ **FreshInputVariable()**
 // Define hypothesis space and model program executions
 $l_f, \{z_i\}_{i=1}^N, A \leftarrow$ **Generate($\mathcal{G}$,$[\![\cdot]\!]$,$P$,$\{I_i\}_{i=1}^N$)**
 // Compute total description length
 $\ell =$ **FreshRealVariable()**
 $A \leftarrow A \cup \{$(= $\ell$ (+ $l_f$ $n(x_1|z_1)$ $n(x_2|z_2)$ $\cdots$ $n(x_N|z_N)$ $p(I_1)$ $p(I_2)$ $\cdots$ $p(I_N)$))$\}$
 // Satisfiability checked by SMT solver
 **while** $A$ satisfiable **do**
   $\sigma \leftarrow$ a satisfying solution to $A$
   $A \leftarrow A \cup \{$(< $\ell$ $\sigma[\ell]$)$\}$
 **end while**
 **let** $f =$ unique string in $\mathcal{L}$ specified by $c$'s of $\sigma$
 **return** $f, \{\sigma[I_i]\}_{i=1}^N, \sigma[\ell]$

---

# 2 Visual concept learning

## 2.1 Generative model

We model visual concept learning in the spirit of *inverse graphics* [4]. The observed data come from a set of images, and our goal here is to learn a model (synthesize a program) that can draw each of those images. We consider a space of programs that invoke simple graphics procedures, and so learning can be thought of as inverting these graphics routines.

Although in principle these graphics routines render down to pixels, we believe that such an unstructured high dimensional input would be handled poorly by any complete constraint solver, such as SMT solvers. These solvers excel in structured, symbolic, and highly constrained domains. So, we implemented a bottom-up image parser that converts images into a structured parse. The synthesized programs output these parses as well, and so both the $\{x_i\}_{i=1}^N$ (observed data) and $\{z_i\}_{i=1}^N$ (program outputs) take the former of parses. Figure 2.1 diagrams this generative model and Figure 3 illustrates an image parse, whose structure we now describe:

The idea behind the parser is to pick out the objects in the image, and identify relations between them, such as whether one object touches another object or whether one object is a smaller version of another one. Concretely, each parse describes $S$ shapes ($S = 2$ in Figure 3) by their coordinates, scale, and *shape ID*. The parser identifies shapes that are identical up to rescaling and assigns them all the same *shape ID* (thus $s_1$ and $s_2$ have distinct *shape ID*). Scale is given relative to the largest shape with the same ID (thus if no shape IDs are duplicated, as in Figure 3, all scales are equal to 1). Following these shape descriptions is a sequence of binary topological relations found to hold among the $S$ shapes. Our parser finds points of contact between shapes (the "borders" relation) and complete containment between shapes (the "contains" relation). Figure 3 has one "borders" relation because $s_1$ and $s_2$ touch each other.

Because we synthesize imperative programs, our model is necessarily directed, that is, causal. However, some of these visual concepts involve undirected constraints relating to the topology of the image. We work around this by (1) having the parser automatically detect these constraints (borders, contains); and (2) allowing the graphics program to print a list of enforced undirected constraints. Some of these printed constraints can be violable, which gives us "optional containment" or "optional bordering" as in Table 1 of the paper. Our noise model $n(\cdot|\cdot)$ specifies how program outputs ($z_i$ in Figure 3) relate to observed parses ($x_i$ in Figure 3), and probabilistically decides whether to enforce optional constraints.

Figure 2: Generative model for visual concept learning. Our parser converts each of the $N$ images into a symbolic parse. This mapping is deterministic and many-to-one. Graphics program $f$ is run on program input $I_i$, yielding another parse $z_i$, which noise model $n(\cdot|\cdot)$ perturbs to produce the $i^{th}$ parse, $x_i$.

Figure 3: We parse images by segmenting shapes and identifying their topological relations (contains, borders), emmitting their coordinates, topological relations, and scales.

## 2.2 Synthesized programs

For each positive and negative example of the SVRT concepts we sampled three example images and synthesize the program shown below as illustrative examples of the systems behavior and performance. Solver timeout was 30sec.

| SVRT Concept | Two example images | | Synthesized program | Synthesis time |
|---|---|---|---|---|
| 1, negative | | | ```(teleport r[0])```<br>```(draw s[0])```<br>```(teleport r[1])```<br>```(draw s[1])``` | < 1 sec |
| 1, positive | | | ```(teleport r[0])```<br>```(draw s[0])```<br>```(teleport r[1])```<br>```(draw s[0])``` | < 1 sec |
| 2, negative | | | ```(teleport r[0])```<br>```(draw s[0])```<br>```(jitter)```<br>```(draw s[1])```<br>```(assert (contains 1 0))``` | < 1 sec |
| 2, positive | | | ```(teleport r[0])```<br>```(draw s[0])```<br>```(teleport r[1])```<br>```(draw s[1])```<br>```(assert (contains 1 0))``` | < 1 sec |
| 3, negative | | | ```(teleport r[0])```<br>```(draw s[0])```<br>```(teleport r[2])```<br>```(draw s[1])```<br>```(teleport r[1])```<br>```(draw s[2])```<br>```(move l[0] 0deg)```<br>```(draw s[3])```<br>```(assert (borders 2 3))```<br>```(assert (borders 0 3))``` | 238 sec |
| 3, positive | | | ```(teleport r[0])```<br>```(draw s[0])```<br>```(teleport r[1])```<br>```(draw s[1])```<br>```(teleport r[2])```<br>```(draw s[2])```<br>```(move l[0] 0deg)```<br>```(draw s[3])```<br>```(assert (borders 1 3))```<br>```(assert (borders 0 2))``` | 217 sec |

| SVRT Concept | Two example images | | Synthesized program | Synthesis time |
|---|---|---|---|---|
| 4, negative |  |  | ```(teleport r[0])```<br>```(draw s[0])```<br>```(teleport r[1])```<br>```(draw s[1])``` | < 1 sec |
| 4, positive |  |  | ```(teleport r[0])```<br>```(draw s[0])```<br>```(teleport r[1])```<br>```(draw s[1])```<br>```(assert (contains 1 0))``` | < 1 sec |
| 5, negative |  |  | ```(teleport r[0])```<br>```(draw s[0])```<br>```(teleport r[2])```<br>```(draw s[1])```<br>```(teleport r[1])```<br>```(draw s[2])```<br>```(move l[0] 0deg)```<br>```(draw s[3])``` | ? sec |
| 5, positive |  |  | ```(teleport r[0])```<br>```(draw s[0])```<br>```(move l[0] 0deg)```<br>```(draw s[0])```<br>```(teleport r[1])```<br>```(draw s[1])```<br>```(teleport r[2])```<br>```(draw s[1])``` | 239 sec |
| 6, negative |  |  | ```(teleport r[0])```<br>```(draw s[0])```<br>```(teleport r[1])```<br>```(draw s[0])```<br>```(teleport r[2])```<br>```(draw s[1])```<br>```(move l[0] 0deg)```<br>```(draw s[1])``` | 228 sec |

| SVRT Concept | Two example images | | Synthesized program | Synthesis time |
|---|---|---|---|---|
| 6, positive |  |  | `(teleport r[0])`<br>`(draw s[0])`<br>`(teleport r[2])`<br>`(draw s[1])`<br>`(move l[0] 0deg)`<br>`(draw s[1])`<br>`(teleport r[1])`<br>`(draw s[0])` | 224 sec |
| 7, negative |  |  | `(teleport r[0])`<br>`(draw s[0])`<br>`(teleport r[5])`<br>`(draw s[1])`<br>`(teleport r[4])`<br>`(draw s[2])`<br>`(teleport r[3])`<br>`(draw s[0])`<br>`(teleport r[2])`<br>`(draw s[2])`<br>`(teleport r[1])`<br>`(draw s[1])` | 794 sec |
| 7, positive |  |  | `(teleport r[0])`<br>`(draw s[0])`<br>`(move l[0] 0deg)`<br>`(draw s[1])`<br>`(teleport r[2])`<br>`(draw s[1])`<br>`(teleport r[3])`<br>`(draw s[1])`<br>`(teleport r[1])`<br>`(draw s[0])`<br>`(teleport r[4])`<br>`(draw s[0])` | 789 sec |
| 8, negative |  |  | `(teleport r[0])`<br>`(draw s[0])`<br>`(teleport r[1])`<br>`(draw s[1])`<br>`(assert`<br>`   (contains-option 1 0))` | < 1 sec |
| 8, positive |  |  | `(teleport r[0])`<br>`(draw s[0] :scale z)`<br>`(jitter)`<br>`(draw s[0])`<br>`(assert (contains 1 0))` | < 1 sec |

| SVRT Concept | Two example images | | Synthesized program | Synthesis time |
|---|---|---|---|---|
| 9, negative |  |  | (teleport r[0])<br>(draw s[0] :scale z)<br>(move l[1] 0deg)<br>(draw s[0] :scale z)<br>(move l[0] 0deg)<br>(draw s[0]) | 33 sec |
| 9, positive |  |  | (teleport r[0])<br>(draw s[0] :scale z)<br>(move l[1] 0deg)<br>(draw s[0] :scale z)<br>(move l[0] 0deg)<br>(draw s[0]) | 41 sec |
| 10, negative |  |  | (teleport r[0])<br>(draw s[0])<br>(teleport r[2])<br>(draw s[0])<br>(teleport r[1])<br>(draw s[0])<br>(move l[0] 0deg)<br>(draw s[0]) | 242 sec |
| 10, positive |  |  | (teleport r[0])<br>(draw s[0])<br>(move l[0] 0deg)<br>(draw s[0])<br>(move l[0] -90deg)<br>(draw s[0])<br>(move l[0] -90deg)<br>(draw s[0]) | 2 sec |
| 11, negative |  |  | (teleport r[0])<br>(draw s[0])<br>(teleport r[1])<br>(draw s[1]) | < 1 sec |
| 11, positive |  |  | (teleport r[0])<br>(draw s[0])<br>(teleport r[1])<br>(draw s[1])<br>(assert (borders 0 1)) | < 1 sec |

| SVRT Concept | Two example images | | Synthesized program | Synthesis time |
|---|---|---|---|---|
| 12, negative |  |  | `(teleport r[0])`<br>`(draw s[0])`<br>`(teleport r[1])`<br>`(draw s[2])`<br>`(move l[0] 0deg)`<br>`(draw s[1])` | 33 sec |
| 12, positive |  |  | `(teleport r[0])`<br>`(draw s[0])`<br>`(move l[1] 0deg)`<br>`(draw s[2])`<br>`(move l[0] -90deg)`<br>`(draw s[1])` | 40 sec |
| 13, negative |  |  | `(teleport r[0])`<br>`(draw s[0])`<br>`(teleport r[2])`<br>`(draw s[0])`<br>`(teleport r[1])`<br>`(draw s[1])`<br>`(move l[0] 0deg)`<br>`(draw s[1])` | 243 sec |
| 13, positive |  |  | `(teleport r[0])`<br>`(draw s[0])`<br>`(move l[0] 0deg)`<br>`(draw s[0])`<br>`(teleport r[1])`<br>`(draw s[1])`<br>`(move l[0] 0deg)`<br>`(draw s[1])` | 64 sec |
| 14, negative |  |  | `(teleport r[0])`<br>`(draw s[0])`<br>`(teleport r[1])`<br>`(draw s[0])`<br>`(move l[0] 0deg)`<br>`(draw s[0])` | 61 sec |

| SVRT Concept | Two example images | | Synthesized program | Synthesis time |
|---|---|---|---|---|
| 14, positive |  |  | ```<br>(teleport r[0])<br>(draw s[0])<br>(move l[1] 0deg)<br>(draw s[0])<br>(move l[0] 0deg)<br>(draw s[0])<br>``` | 33 sec |
| 15, negative |  |  | ```<br>(teleport r[0])<br>(draw s[0])<br>(move l[0] 0deg)<br>(draw s[2])<br>(move l[0] -90deg)<br>(draw s[3])<br>(move l[0] -90deg)<br>(draw s[1])<br>``` | 2 sec |
| 15, positive |  |  | ```<br>(teleport r[0])<br>(draw s[0])<br>(move l[0] 0deg)<br>(draw s[0])<br>(move l[0] 90deg)<br>(draw s[0])<br>(move l[0] 90deg)<br>(draw s[0])<br>``` | 2 sec |
| 16, negative |  |  | ```<br>(teleport r[0])<br>(draw s[0])<br>(flip-X)<br>(draw s[0])<br>(teleport r[1])<br>(draw s[0])<br>(flip-X)<br>(draw s[0])<br>(teleport r[2])<br>(draw s[0])<br>(flip-X)<br>(draw s[0])<br>``` | 69 sec |
| 16, positive |  |  | ```<br>(teleport r[0])<br>(draw s[0])<br>(flip-X)<br>(draw s[1])<br>(teleport r[2])<br>(draw s[0])<br>(flip-X)<br>(draw s[1])<br>(teleport r[1])<br>(draw s[0])<br>(flip-X)<br>(draw s[1])<br>``` | 86 sec |

| SVRT Concept | Two example images | | Synthesized program | Synthesis time |
|---|---|---|---|---|
| 17, negative |  |  | (teleport r[0])<br>(draw s[0])<br>(teleport r[1])<br>(draw s[0])<br>(teleport r[2])<br>(draw s[1])<br>(move l[0] 0deg)<br>(draw s[0]) | 230 sec |
| 17, positive |  |  | (teleport r[0])<br>(draw s[0])<br>(teleport r[1])<br>(draw s[0])<br>(teleport r[2])<br>(draw s[0])<br>(move l[0] 0deg)<br>(draw s[1]) | 243 sec |
| 18, negative |  |  | (teleport r[0])<br>(draw s[0])<br>(move l[0] 0deg)<br>(draw s[0])<br>(teleport r[1])<br>(draw s[0])<br>(move l[0] 0deg)<br>(draw s[0])<br>(teleport r[3])<br>(draw s[0])<br>(teleport r[2])<br>(draw s[0]) | 541 sec |
| 18, positive |  |  | (teleport r[0])<br>(draw s[0])<br>(flip-X)<br>(draw s[0])<br>(teleport r[2])<br>(draw s[0])<br>(flip-X)<br>(draw s[0])<br>(teleport r[1])<br>(draw s[0])<br>(flip-X)<br>(draw s[0]) | 781 sec |
| 19, negative |  |  | (teleport r[0])<br>(draw s[0])<br>(teleport r[1])<br>(draw s[1]) | < 1 sec |

| SVRT Concept | Two example images | | | Synthesized program | Synthesis time |
|---|---|---|---|---|---|
| 19, positive |  |  | | `(teleport r[0])`<br>`(draw s[0] :scale z)`<br>`(teleport r[1])`<br>`(draw s[0])` | < 1 sec |
| 20, negative |  |  | | `(teleport r[0])`<br>`(draw s[0])`<br>`(teleport r[1])`<br>`(draw s[1])` | < 1 sec |
| 20, positive |  |  | | `(teleport r[0])`<br>`(draw s[0])`<br>`(teleport r[1])`<br>`(draw s[1])` | < 1 sec |
| 21, negative |  |  | | `(teleport r[0])`<br>`(draw s[0])`<br>`(teleport r[1])`<br>`(draw s[1])` | < 1 sec |
| 21, positive |  |  | | `(teleport r[0])`<br>`(draw s[0])`<br>`(teleport r[1])`<br>`(draw s[1])` | < 1 sec |
| 22, negative |  |  | | `(teleport r[0])`<br>`(draw s[0])`<br>`(move l[1] 0deg)`<br>`(draw s[2])`<br>`(move l[0] 0deg)`<br>`(draw s[1])` | 34 sec |

| SVRT Concept | Two example images | | Synthesized program | Synthesis time |
|---|---|---|---|---|
| 22, positive | | | `(teleport r[0])`<br>`(draw s[0])`<br>`(move l[1] 0deg)`<br>`(draw s[0])`<br>`(move l[0] 0deg)`<br>`(draw s[0])` | 37 sec |
| 23, negative | | | `(teleport r[0])`<br>`(draw s[0])`<br>`(teleport r[1])`<br>`(draw s[1])`<br>`(jitter)`<br>`(draw s[2])`<br>`(assert (contains 2 1))`<br>`(assert (contains 2 0))` | < 1 sec |
| 23, positive | | | `(teleport r[0])`<br>`(draw s[0])`<br>`(teleport r[1])`<br>`(draw s[1])`<br>`(move l[0] 0deg)`<br>`(draw s[2])`<br>`(assert (contains 2 0))` | 33 sec |

## 2.3 Program denotations

Vision programs control a turtle [5], and we represent the state of the program as a tuple of the turtle's $x$, $y$ coordinates, and its orientation $\theta$ as a tuple of $\cos\theta$, $\sin\theta$. The input to the program is a function $I$ that maps the names of latent variables (shapes, distances, positions, angles, scales, and jitter) to their values. For example, $I(\theta_0)$ is an angle input to the program. The variable $\sigma$ ranges over states of the turtle: $\sigma = \langle x, y, \cos\theta, \sin\theta \rangle$. In what follows we will slightly abuse notation and freely mix SMT terms with the equations they represent (eg, writing $x + y$ for `(+ x y)`) whenever doing so introduces no ambiguity.

We define a function `rotate` that takes as input an orientation $\langle \cos\theta, \sin\theta \rangle$ and an angle of rotation $\langle \cos\alpha, \sin\alpha \rangle$, returning the new orientation after rotation by angle $\alpha$:

$$\texttt{rotate}(\langle \cos\theta, \sin\theta \rangle, \langle \cos\alpha, \sin\alpha \rangle) = \langle \cos\theta\cos\alpha - \sin\theta\sin\alpha, \cos\theta\sin\alpha + \sin\theta\cos\alpha \rangle \qquad (7)$$

The denotations of each production in Table 1 are

$$\llbracket \text{teleport}(\mathcal{R}, \theta_0) \rrbracket(\sigma, I) = \langle \llbracket \mathcal{R} \rrbracket(I), I(\theta_0) \rangle \tag{8}$$

$$\llbracket \text{flipX}() \rrbracket(\langle x, y, \cos\theta, \sin\theta \rangle, I) = \langle -x, y, \cos\theta, \sin\theta \rangle \tag{9}$$

$$\llbracket \text{flipY}() \rrbracket(\langle x, y, \cos\theta, \sin\theta \rangle, I) = \langle x, -y, \cos\theta, \sin\theta \rangle \tag{10}$$

$$\llbracket \text{jitter}() \rrbracket(\langle x, y, \cos\theta, \sin\theta \rangle, I) = \langle x + I(j_x), y + I(j_y), \cos\theta, \sin\theta \rangle \tag{11}$$

$$\llbracket r_j \rrbracket(I) = I(r_j) \tag{12}$$

$$\llbracket \theta_j \rrbracket(\langle x, y, \cos\theta, \sin\theta \rangle, I) = \texttt{rotate}(\langle \cos\theta, \sin\theta \rangle, I(\theta_j)) \tag{13}$$

$$\llbracket 0° \rrbracket(\langle x, y, \cos\theta, \sin\theta \rangle, I) = \langle \cos\theta, \sin\theta \rangle \tag{14}$$

$$\llbracket 90° \rrbracket(\langle x, y, \cos\theta, \sin\theta \rangle, I) = \langle -\sin\theta, \cos\theta \rangle \tag{15}$$

$$\llbracket -90° \rrbracket(\langle x, y, \cos\theta, \sin\theta \rangle, I) = \langle \cos\theta, -\sin\theta \rangle \tag{16}$$

$$\llbracket \ell_j \rrbracket(I) = I(\ell_j) \tag{17}$$

$$\llbracket z_j \rrbracket(I) = I(z_j) \tag{18}$$

$$\llbracket 1 \rrbracket(I) = 1 \tag{19}$$

$$\llbracket s_j \rrbracket(I) = I(s_j) \tag{20}$$

$$\llbracket \text{draw}(\mathcal{S}, \mathcal{Z}) \rrbracket(\langle x, y, \cos\theta, \sin\theta \rangle, I) = \langle x, y, s_0, s_1 \times \llbracket \mathcal{Z} \rrbracket(I) \rangle \tag{21}$$

$$\text{where} \quad \langle s_0, s_1 \rangle = \llbracket \mathcal{S} \rrbracket(I) \tag{22}$$

$$\llbracket \text{contains}(j, k) \rrbracket(\sigma, I) = \langle \sigma, \text{contains}_{j,k}, \text{True} \rangle \tag{23}$$

$$\llbracket \text{borders}(j, k) \rrbracket(\sigma, I) = \langle \sigma, \text{borders}_{j,k}, \text{True} \rangle \tag{24}$$

$$\llbracket \text{contains?}(j, k) \rrbracket(\sigma, I) = \langle \sigma, \text{contains}_{j,k}, \text{False} \rangle \tag{25}$$

$$\llbracket \text{borders?}(j, k) \rrbracket(\sigma, I) = \langle \sigma, \text{borders}_{j,k}, \text{False} \rangle \tag{26}$$

$$\llbracket \text{move}(\mathcal{L}, \mathcal{A}) \rrbracket(\langle x, y, \cos\theta, \sin\theta \rangle, I) = \langle x + \llbracket \mathcal{L} \rrbracket(I)\cos\alpha, y + \llbracket \mathcal{L} \rrbracket(I)\sin\alpha, \cos\alpha, \sin\alpha \rangle \tag{27}$$

We can remove the nonlinearities in the SMT problem if there is at most one input length and at most one input angle (the initial angle), in which case we use the alternative denotations

$$\llbracket \text{move}(\mathcal{L}, \mathcal{A}) \rrbracket(\sigma, I) = \llbracket \mathcal{A} \rrbracket(\sigma) \tag{28}$$

$$\llbracket 0° \rrbracket(\langle x, y, \cos\theta, \sin\theta \rangle) = \langle x + \cos\theta, y + \sin\theta, \cos\theta, \sin\theta \rangle \tag{29}$$

$$\llbracket 90° \rrbracket(\langle x, y, \cos\theta, \sin\theta \rangle) = \langle x - \sin\theta, y + \cos\theta, -\sin\theta, \cos\theta \rangle \tag{30}$$

$$\llbracket -90° \rrbracket(\langle x, y, \cos\theta, \sin\theta \rangle) = \langle x + \sin\theta, y - \cos\theta, \sin\theta, -\cos\theta \rangle \tag{31}$$

We now specify the SMT encoding of the noise model. Let the program output $z$ consist of $N$ shapes with coordinates $\{(x_i, y_i)\}_{i=1}^N$, shape IDs $\{s_i\}_{i=1}^N$, and scales $\{z_i\}_{i=1}^N$. Let $z$ also have $B$ (resp. $C$) borders (resp. contains) relations $\{\langle j^i, k^i, b^i \rangle\}_{i=1}^B$ (resp. $\{\langle j^i, k^i, c^i \rangle\}_{i=1}^C$). Let the observation $x$ consist of $N$ shapes with coordinates $\{(x_i', y_i')\}_{i=1}^N$, shape IDs $\{s_i'\}_{i=1}^N$, and scales $\{z_i'\}_{i=1}^N$. Let $x$ also have $B'$ (resp. $C'$) borders (resp. contains) relations $\{\langle j'^i, k'^i \rangle\}_{i=1}^{B'}$ (resp. $\{\langle j'^i, k'^i \rangle\}_{i=1}^{C'}$). Then

$$n(x|z) = -\log P_{x|z}(x|z) \overset{+}{=} \log 2 \times \sum_{i=1}^B \mathbb{1}[\neg b^i] + \log 2 \times \sum_{i=1}^C \mathbb{1}[\neg c^i] \tag{32}$$

whenever

$$
\begin{align}
\forall i : s_{\pi(i)} &= s'_i \tag{33}\\
\forall i : z_{\pi(i)} &= z'_i \tag{34}\\
\forall i : |x_{\pi(i)} - x'_i| &< \delta = 3 \tag{35}\\
\forall i : |y_{\pi(i)} - y'_i| &< \delta = 3 \tag{36}\\
\forall i : c^i \Rightarrow &\bigvee_{i'} (\pi(j^i) = j'^{i'} \wedge \pi(k^i) = k'^{i'}) \tag{37}\\
\forall i' : &\bigvee_{i} (\pi(j^i) = j'^{i'} \wedge \pi(k^i) = k'^{i'}) \tag{38}\\
\forall i : b^i \Rightarrow &\bigvee_{i'} (\pi(j^i) = j'^{i'} \wedge \pi(k^i) = k'^{i'}) \vee (\pi(j^i) = k'^{i'} \wedge \pi(k^i) = j'^{i'}) \tag{39}\\
\forall i' : &\bigvee_{i} (\pi(j^i) = j'^{i'} \wedge \pi(k^i) = k'^{i'}) \vee (\pi(j^i) = k'^{i'} \wedge \pi(k^i) = j'^{i'}) \tag{40}
\end{align}
$$

and $-\log P_{x|z}(x|z) = \infty$ otherwise, where $\pi(\cdot)$ is a permutation of $\{1, \cdots, N\}$, which we encode as $N^2$ binary indicator variables. The extra disjunctions in Equation 39 and Equation 40 compared to Equation 37 and Equation 38 comes from the reflexivity of the borders relation.

We put uniform priors over the parameters of the program and defined the description lengths below. These priors are improper and capture the intuition that each additional continuous degree of freedom imposes a constant description length penalty; one could imagine using the BIC or AIC, but for only a few example images the difference would be small. We arbitrarily put the description length of a shape at a large constant.

$$
\begin{align}
-\log P(s_j) &= 1000 \tag{41}\\
-\log P(r_j) &= 20 \tag{42}\\
-\log P(\ell_j) &= 10 \tag{43}\\
-\log P(\theta_j) &= 10,\ \theta \in [0, 2\pi] \tag{44}\\
-\log P(z_j) &= 10,\ z_j \in [0, 1] \tag{45}\\
-\log P(j_x) = -\log P(j_y) &= 10,\ j_x \in [-7, 7],\ j_y \in [-7, 7] \tag{46}
\end{align}
$$

These decisions correspond to the $p(\cdot)$ of Algorithm 2 in the following way:

$$
\begin{aligned}
p(I) = (\texttt{+}\ &-\log P(I(s_1)) - \log P(I(s_2)) \cdots\\
&-\log P(I(r_1)) - \log P(I(r_2)) \cdots\\
&-\log P(I(\ell_1)) - \log P(I(\ell_2)) \cdots\\
&-\log P(I(\theta_1)) - \log P(I(\theta_2)) \cdots\\
&-\log P(I(z_1)) - \log P(I(z_2)) \cdots\\
&-\log P(I(j_x)) - \log P(I(j_y))\ )
\end{aligned}
$$

# 3  Morphological rule learning

## 3.1  Generative model

We assume the learner observes triples of $\langle$lexeme, tense, word$\rangle$[1]. We can think of these triples as the entries of a matrix whose columns correspond to different tenses and whose rows correspond to different lexemes; see Table 3 of the paper. Each row of this matrix should be thought of as a program output, and the inputs to the program take the form of (unobserved) stems. Figure 4 diagrams these modeling assumptions. Because

Figure 4: Generative model for morphological rule learning. We observe a matrix of inflected forms, $x_{i,t}$, of $T$ inflections and $N$ lexemes. We synthesize program $f = \langle f_1, \cdots, f_T \rangle$ while jointly inferring program inputs $I_i$, which are the underlying stems. A noise model produces $x_{i,t}$ from $z_{i,t}$ which accommodates exceptions. The tuple $z_i \triangleq \langle z_{i,1}, \cdots, z_{i,T} \rangle$ is the output of program $f$.

linguistic rules sometimes have exceptions (for example, the past tense of *run* is *ran*, not *runned*), our noise model assumes that with some small probability $\epsilon$ a matrix entry is produced not from the learned program but instead drawn from a description-length prior over sequences of phonemes.

## 3.2  Program denotations

Let $L$ be the maximum length of any observed sequence of phonemes. The input $I$ to the program is an unobserved stem, which we represent as a tuple of its length $\ell$ and up to $L$ phonemes $p_j$. So $I$ is of the form $\langle \ell, p_1, \cdots, p_L \rangle$.

We will write some denotations subject to certain constraints (eg, writing $[\![x + y]\!] = z$ s.t. $z = [\![x]\!] + [\![y]\!]$). This is meant as a shorthand for a denotation that returns a 2-tuple of a return value and predicate s.t. the return value is correctly computed whenever the predicate holds (eg, $[\![x + y]\!] = \langle z, z = [\![x]\!] + [\![y]\!] \rangle$). We can roll these constraints into an existing noise model $n(\cdot|\cdot)$ by defining a new noise model $n'(\cdot|\cdot)$ where

$$n'(x| \langle z, k \rangle) = \begin{cases} n(x|z) & \text{if } k \\ -\infty & \text{otherwise.} \end{cases} \tag{47}$$

We define a function `append` that concatenates two sequences of phonemes:

$$\texttt{append}(\langle \ell_a, a_1, \cdots, a_L \rangle, \langle \ell_b, b_1, \cdots, b_L \rangle) = \langle \ell_c, c_1, \cdots, c_L \rangle \tag{48}$$

s.t.

$$\ell_c = \ell_a + \ell_b \tag{49}$$
$$\forall 0 \le j \le L : \ell_a > j \Longrightarrow c_j = a_j \tag{50}$$
$$\forall 0 \le j \le L : \ell_a = j \Longrightarrow \forall 0 \le i \le L : \ell_b > i \Longrightarrow c_{j+i} = b_i \tag{51}$$

and a function `last` that extracts the end of a sequence of phonemes:

$$\texttt{last}(\langle \ell, p_1, \cdots, p_L \rangle) = p \tag{52}$$

s.t.

$$\forall 1 \le j \le L : \ell = j \Longrightarrow p = p_j \tag{53}$$

The denotations of each production in Table 2 of the paper are

$$[\![\langle \mathcal{C}, \cdots, \mathcal{C} \rangle]\!](I) = \langle [\![\mathcal{C}]\!](I), \cdots, [\![\mathcal{C}]\!](I) \rangle \tag{54}$$

$$[\![\text{if } (\mathcal{G}) \ \mathcal{R} \text{ else } \mathcal{C}]\!](I) = \text{if}([\![\mathcal{G}]\!](I)), [\![\mathcal{R}]\!](I), [\![\mathcal{C}]\!](I)) \tag{55}$$

$$[\![\text{stem} + \text{phoneme}^*]\!](I) = \texttt{append}([\![\text{stem}]\!](I), [\![\text{phoneme}^*]\!](I)) \tag{56}$$

$$[\![\mathcal{VPMS}]\!](I) = [\![\mathcal{V}]\!](I) \wedge [\![\mathcal{P}]\!](I) \wedge [\![\mathcal{M}]\!](I) \wedge [\![\mathcal{S}]\!](I) \tag{57}$$

$$[\![?]\!](I) = \text{True} \tag{58}$$

$$[\![\text{stem}]\!](I) = I \tag{59}$$

$$[\![p \in P]\!](I) = p, \ P \text{ is the set of all phonemes} \tag{60}$$

along with denotations for each phonetic feature which serve to check if the end of the stem has that feature. For example,

$$[\![\text{VOICED}]\!](I) = \texttt{last}(I) \in \{\text{d}, \text{ə}, \cdots\} \tag{61}$$

Our system has access to the following phonetic features: place (labial, coronal, or dorsal); manner (stop, fricative, nasal, liquid, glide); voicing (voiced, unvoiced); sibilancy (sibilant, not sibilant). The phoneme/feature matrix is drawn from a standard linguistics textbook [6] and pronunciations are drawn from CMUDict.[2]

Both observations $x$ and program outputs $z$ consist of a 5-tuple of words: one for the present, past, progressive, past participle, and 3rd person singular present. Let $x = \langle w_x^1, \cdots, w_x^5 \rangle$ and $z = \langle w_z^1, \cdots, w_z^5 \rangle$. Then we encode our noise model as an SMT term by upper bounding the description length as

$$-\log P_{x|z}(\langle w_x^1, \cdots, w_x^5 \rangle \mid \langle w_z^1, \cdots, w_z^5 \rangle) \leq \sum_{t=1}^{5} \begin{cases} -\log(1 - \epsilon), \text{ if } w_x^t = w_z^t \\ -\log L\epsilon - |w_x^t| \log |P|, \text{ else} \end{cases} = n(x|z) \tag{62}$$

where $P$ is the set of phonemes ($|P| = 44$) and $\epsilon = 0.1$ in our experiments.

The description length prior over program inputs $I = \langle \ell, p_1, ..., p_L \rangle$ is encoded as an SMT term as

$$p(I) = \log P_I(\langle \ell, p_1, ..., p_L \rangle) = \log L + \ell \log |P| \tag{63}$$

## 3.3   Solver times for morphological rule learner

We fit our models using random sample consensus (RANSAC) [7] to allow training on large data sets. We observed dramatic slowdowns as the number of lexemes increased (see Fig. 5). The size of the resulting SMT problem grows linearly in the training data, and the problem variables are all coupled through the description length calculation, which heuristically accounts for the observed slowdown. Two areas of further research are (1) better RANSAC-inspired techniques, like CEGIS [1], for applying solvers to noisy data, and (2) nonsolver based techniques for unsupervised program synthesis, like stochastic search.

# 4   Neural network training

We consider two convolutional neural network architectures:

- A variant of LeNet5 [8]. We used two convolutional layers followed by a fully connected hidden layer. The network took as input one grayscale $28 \times 28$ input plane, while the first convolutional layer produced 20 output planes and the second convolutional layer produced 50 output planes. Convolutional kernels were $5 \times 5$ and were followed by rectified linear units and a max pooling operation with a window size of 2. We used 500 hidden units followed by rectified linear units, whose output went to two soft max units. This network solved 6/23 SVRT problems with an average classification accuracy of 70.0% and a correlation with the human data of $r = 0.48$.

Figure 5: Average solver time as a function of random sample size. Log base $e$. Each lexeme had five inflections provided.

- A variant of AlexNet [9]. To combat over fitting, we removed two layers of convolution and two fully connected layers. The network took as input one grayscale $128 \times 128$ input plane. The first convolutional layer used $11 \times 11$ kernels with a step size of 4 and produced 96 output planes. The second convolutional layer used $5 \times 5$ kernels with a step size of 1 and produced 256 output planes. The third convolutional layer used $3 \times 3$ kernels with a step size of 1 and produced 384 output planes. Each convolutional layer was followed by rectified linear units and a max pooling operation with a window size of 2. The 384-dimensional output of the last max pooling layer went to two linear units followed by a soft max operation. This network solved 3/23 SVRT problems with an average classification accuracy of 64.9% and a correlation with the human data of $r = 0.55$.

Networks were trained using stochastic gradient descent for 300 epochs with a batch size of 100 and a learning rate of 0.05. We used negative log likelihood as our loss function.

## Footnotes

[1]The lexeme is the meaning of the stem or root; for example, *run*, *ran*, *runs* all share the same lexeme

[2]`www.speech.cs.cmu.edu/cgi-bin/cmudict`