[Reviews · NeurIPS 2015]

Submitted by Assigned_Reviewer_1

This paper works on the idea of explaining a set of data points x_i i = 1..N in terms of finding a program (drawn from a prior over programs, and common for all i = 1..N), and specific inputs I_i (i=1..N) for each datapoint. The program is built up out of domain specific primitives, as shown in Tables 1 and 2. The method is applied

to visual concept learning and morphological rule learning tasks which have a cognitive science flavor.

Of course this idea is quite similar to standard latent variable modelling. However, rather than say having a matrix of parameters (as

in factor analysis) or cluster centers (as in clustering) which define the model, we are allowed here a more general program to map from

I to x, see e.g. Fig 1 (right). However, the program can make use of

explicit domain-specific prior knowledge, as per Tables 1 and 2.

For sec 2 see comments under

Clarity.

In sec 3.1 the method is applied to data from the SVRT (Synthetic Visual Reasoning Task), as per Fig 1. The evaluation as per line 203 is quite hard to understand, but I made sense of if as comparing P_x(t|E_j) for models based on different sets of examples j = 1,2. The paper has an inadequate discussion of the results -- based on raw accuracy (dashed lines in Fig 3) it seems that the program synthesis not doing much better than "image features" (0.78 vs 0.75)? But was the "image features" method trained with far more examples? The r values quoted assess comparison with human accuracy, but what is the goal of the paper -- unsupervised learning, or cognitive modeling?

In sec 3.2 the method is applied to morphological rule learning. Again the details are very compressed -- words are represented by sequences of phonemes.

I have no idea from the description how features like VOICED are defined -- e.g. do they apply to phonemes, or the word as a whole?

Clarity: I found the paper rather lacking in detail -- this is likely due to there being a lot of ideas that are brought together. It is clearly hard to fit these into a NIPS-length paper, but it would have been possible to use the supp mat to expand on various issues and make them intelligible.

For example Alg 1 (SMT encoding of expressions) is impenetrable. K(N', N'' ..) is undefined. This whole thing needs much more explanation; it remains very unclear to me what the basic idea of SMT solving is here and how it works. A worked example (e.g. in supp mat) would help a lot. Minor: What is the meaning of the if() function used in Alg 1?

See also comments above about level of detail in ecs 3.1 and 3.2. Given these issues I suggest that the authors write a long paper which they make available as supp mat -- in this case the NIPS paper would really be a summary of that long paper.

The statements about key ideas in l 422-424 could usefully be moved up to the introduction.

Originality: Wrt the key ideas mentioned in l 422-424, the point about a domain specific grammar seems to be a common idea, e.g.

"background knowledge" in ILP. The "encoding of soft probabilistic

constraints as hard constraints for symbolic search" may be more novel, but the brevity of sec 2.2 makes it hard to understand this in detail.

Significance: This paper is tackling a hard and interesting problem, that of the unsupervised learning of programs; this generalizes unsupervised learning applied to vector data to other kinds of data (e.g. sequences).

Overall: Part of the difficulty in understanding this paper is that it is actually covering a lot of ground, and it is hard to fit this into the NIPS format. I have no knowledge about SMT, but if the

authors' claim about novelty wrt "encoding of soft probabilistic

constraints as hard constraints for symbolic search" is correct, then it seems to me that the paper is making a useful advance in a hard area. But it would be really helpful if the authors could produce

a significantly longer version where there is actually space to

explain what is going on.

Minor:

l 098 \Sigma -> \tau -> \tau ??

N in Alg 1 contradicts its defn in l 077.

Fig 3 -- the different colors are unnecessary, and the use of

yellow makes the ConvNet plot hard to read.

l 337 I happen to know about RANSAC, but many readers won't. It would be useful to explain that RANSAC uses a minimal set of examples to learn a model, and then evaluate it on all of the available examples.

ADDED AFTER REBUTTAL: =====================

The other reviews and the author response have been read.

I retain my score of 7, but strongly encourage the authors to make

extensive supplementary material available; at the moment the paper is more like an "advert" (Donoho's phrase), and we need much more detail if others are able to reproduce the work. In para 1 of the

rebuttal the authoros say "we could include [...] an example of an unsupervised program synthesis problem being converted into an SMT

formula". The "could" here is not enough -- we need to see more detail.
Summary: This paper works on the idea of explaining a set of data points x_i i = 1..N in terms of finding a program (drawn from a prior over programs, and common for all i = 1..N), and specific inputs I_i (i=1..N) for each datapoint. The program is built up out of domain specific primitives, as shown in Tables 1 and 2. The method is applied to visual concept learning and morphological rule learning tasks which have a cognitive science flavor. If the authors' claim about novelty wrt "encoding of soft probabilistic constraints as hard constraints for symbolic search" is correct, then it seems to me that the paper is making a useful advance in a hard area.

Submitted by Assigned_Reviewer_2

Unsupervised Learning by Program Synthesis" shows how to use search algorithms over programs (expression trees) and program inputs to find short explanatory programs of data sets and their individual examples respectively.

The paper may well be novel in one or more ways, but it was not clear to me

what the contribution(s) of this paper was(were), because the paper not did

make the existing state of the art clear enough. The paper has too much of a

"this is a thing you could do" flavor to it, and not enough "relative to what you are doing, you might consider doing something else because ___". The paper does seem to be opening a new application area, so it should strive to connect to existing application areas.

The algorithms used to search a grammar for possible explanations don't seem to be the focus here, nor the algorithms for searching for possible inputs

to make a particular program explain a particular example (image or (lexeme,

tense, word) tuple). Instead a general SMT solver is doing the heavy lifting.

Expression (1) looks like a likelihood, and the approach is introduced as a

strategy for unsupervised learning, but there is no mention of likelihood in

the empirical section.

Although I will admit this paper does not fall squarely into my area of

expertise, it seems to me that what's proposed is a learning algorithm (or

arguably a compression algorithm) for a new model class.

There are existing

algorithms and metrics for judging progress in learning and compression, and

this paper needs to use those metrics. I would be interested in, a comparison with the techniques used in [1] for example.

Another area of improvement for this paper would be presentation clarity. It is not clear to me how exactly the pictures in e.g. Figure 1 are modeled. Eq 1 suggests that we'll be modeling the image, but the math around Equation 5 seems dedicated to classification, rather than unsupervised learning. What's going on? Also the approximation sum->max in Equation 5 seems like it could be pretty severe, especially when it's used to quantify the effect of adding a single example to E1 or E2.

[1] http://www.cs.toronto.edu/~rgrosse/phd-thesis.pdf
Summary: The paper may well be novel in one or more ways, but it was not clear to me

what the contribution(s) of this paper was(were), because the paper not did

make the existing state of the art clear enough.

Submitted by Assigned_Reviewer_3

(I have read the author response and it did not make me change my review.)
Summary: This paper approaches unsupervised learning from a data set as a problem of minimizing the joint description length of a program and its input that together produce the observed data. The paper solves the problem by applying program synthesis (by sketching and SMT solving, with their associated limitations). The approach was able to learn and classify visual concepts from examples, and to learn the basics of English verb inflections.

Submitted by Assigned_Reviewer_4

Some part of the actual implement seem to be less well explained to me at first, but are clarified in the authors responses. I have read considered the rebuttal.
Summary: This paper proposes to leverage unsupervised learning (compression) for concept learning and prediction tasks. The overall formalism (equation 1) is quite intuitive and reasonable. As far as I know, this is the first to put program synthesis (a challenging task by itself) in a beautiful machine learning formalism. It is a bit disappointing though that program induction is mainly handled by domain knowledge in the form of production grammars. Since program induction is usually the most expensive step in program synthesis.

Author Feedback
Author rebuttal: Regarding all reviews:
As multiple reviewers note, the details of the synthesis algorithm, particularly the SMT encoding, are very compressed. Our intention is that a reader familiar with the program synthesis literature will be able to understand the technical details, and that a reader familiar only with the AI/ML literature will be able to understand the high-level idea, namely, modeling the execution of a program in a constraint solver (SMT) and asking the constraint solver to solve for the program. In a revised version, we could include, in the supplement, an example of an unsupervised program synthesis problem being converted in to an SMT formula.

Regarding review 1:
For the visual concepts, program synthesis may not seem to be doing much better than "image features", but "image features" was trained on 10000 examples and program synthesis was trained on 6 examples, as stated in the caption to Figure 3.
Phonological features apply to individual phonemes, and we could include in the supplement to a revised version our phoneme/feature matrix.
l 127 defines K(...)

Regarding review 2:
We build on the existing state of the art for program synthesis as follows: we solve jointly for both the program and it's inputs (prior work had treated the inputs as given), and we show how to encode probability distributions over programs in a format amenable to (SMT) solvers, as reviewer 1 notes.
The heavy lifting is done by a combination of a domain-specific grammar, a domain-general algorithm (Alg. 1) for translating these grammars to SMT formulas, and the black-box optimization routine (SMT solver).
Equation (1) is a description length, not a likelihood. Reviewer 3 comments on our combining of program synthesis and machine learning, and equation (1) is central to this.
Roger Grosse's cited thesis is concerned with learning good matrix decompositions for real-valued data. Our work is different in that we are not concerned with learning from real-valued vectors, but from tuples, strings, parses, and other symbolic structures, as reviewer 1 notes.
We are claiming a new way to do unsupervised learning, and therefore we compare to other learning algorithms. It is true that our algorithm might do well on some kinds of compression problems, but that is not our main focus in this work. In the vision domain, we compare with supervised classification algorithms to show that the algorithm can also be used for prediction.

Regarding review 3:
The parse of a graph (SVRT picture) is not read from the oracle program which generates the graph, but from the raw pixels.
Because the program learning is done from an image parse, we did a control experiment where we trained a classifier only on features of the image parse, as detailed in section 3.1.
A noise model is needed because parsing is not perfectly reliable.

Regarding review 6:
wrt baseline comparisons for vision: we tackled a suite of problems (the SVRT) designed to be difficult for conventional machine vision approaches, a fact which we validated empirically. Data sets such as mnist or imagenet would be inappropriate for judging the claims we make about our algorithm, which is designed to learn from small amounts of structured data, rather than large amounts of real-valued data.
wrt baseline comparisons for morphology: we compare with the best comparable stemmer (Morfessor), which did very well in the 2010 Morpho Challenge. Our system is designed to learn from the more structured data one might encounter in, for example, a linguistics problem set, and so we compared with the state-of-the-art in modeling English verb inflections: Albright&Hayes' minimal generalization learner.
To be clear, we do not claim that existing AI systems don't learn good priors over time. We hypothesize that future work could explore the problem of learning sketches over programs, which could be integrated with solver-based synthesis techniques.